# Phytofabrication of Nanoparticles as Novel Drugs for Anticancer Applications

**DOI:** 10.3390/molecules24234246

**Published:** 2019-11-21

**Authors:** Qi-Yao Wei, Kai-Ming He, Jin-Ling Chen, Yan-Ming Xu, Andy T. Y. Lau

**Affiliations:** Laboratory of Cancer Biology and Epigenetics, Department of Cell Biology and Genetics, Shantou University Medical College, Shantou, Guangdong 515041, China

**Keywords:** phytochemicals, phytofabrication, nanoparticles, nanophytochemicals, anticancer

## Abstract

Cancer is one of the foremost causes of death globally and also the major stumbling block of increasing life expectancy. Although the primary treatment of surgical resection, chemotherapy, and radiotherapy have greatly reduced the mortality of cancer, the survival rate is still low because of the metastasis of tumor, a range of adverse drug reactions, and drug resistance. For all this, it is relevant to mention that a growing amount of research has shown the anticarcinogenic effect of phytochemicals which can modulate the molecular pathways and cellular events include apoptosis, cell proliferation, migration, and invasion. However, their pharmacological potential is hindered by their low water solubility, low stability, poor absorption, and rapid metabolism. In this scenario, the development of nanotechnology has created novel formulations to maximize the potential use of phytochemicals in anticancer treatment. Nanocarriers can enhance the solubility and stability of phytochemicals, prolong their half-life in blood and even achieve site-targeting delivery. This review summarizes the advances in utilizing nanoparticles in cancer therapy. In particular, we introduce several applications of nanoparticles combined with apigenin, resveratrol, curcumin, epigallocatechin-3-gallate, 6-gingerol, and quercetin in cancer treatment.

## 1. Introduction

Cancer is nowadays the second leading cause of death, following heart diseases and affecting people of all ages. The conventional treatment methods against cancer have included surgical resection, radiotherapy, chemotherapy, and a combination of any of these treatments [1]. Among them, conventional chemotherapeutics is still the main treatment for many cancers in advanced stages [2]. However, there are challenges associated with the treatment—systemic toxicity, low selectivity, various adverse reactions, etc. Due to the fact that cancer treatment usually uses compounds that target fast-dividing cells, it has untoward side effects on normal, fast-dividing cells such as hair follicles and epithelial cells in the digestive system. Furthermore, one of the aggravating circumstances is that many cancer cells gradually develop resistance to conventional forms of therapy. To reach the best possible therapeutic response, novel drugs and delivery strategies need to be designed.

There is a growing body of convincing evidence suggesting that phytochemical components in food exert anticancer activities in different types of cancer [3]. Phytochemicals, a class of bioactive molecules that can be obtained from fruits, vegetables, grains, and other plant parts, have been proven to be suitable candidates for such a therapy. Numerous studies have demonstrated that these chemopreventive phytochemicals can regulate the cellular and molecular events including apoptosis, cell proliferation, cell cycle, DNA repair, differentiation, and metastasis [3,4,5]. Additionally, many of these natural products are generally less toxic and well tolerated in normal cells. Compared with chemotherapeutic drugs, natural products are tolerable in normal cells even at high doses. Despite tremendous efforts in preclinical settings, the applicability of phytochemicals to humans has met with only limited success. Among the many reasons, inefficient delivery of promising natural agents to the target site could likely be the main reason for the clinical failure. This can be related to poor water solubility of the agents and their chemical instability in the biological environment. Furthermore, poor pharmacokinetic characteristics of phenolic compounds arising from rapid metabolism, poor solubility and stability pose different challenges of toxicity, inefficacy, and low tissue distribution. Thus, the development of novel and effective delivery systems that can ameliorate these setbacks is imperative for cancer treatment.

Prior studies on nanotechnology-based drug delivery systems used for delivering natural agents have suggested to us that this technology may have considerable advantages over conventional therapies for cancer. One advantage of this nanotechnology is that drugs encapsulated in nanoparticles can be protected from destructive action of external media [6]. Thus, the half-life of the drug in the systemic circulation can be prolonged. In addition, it is now well established that nanoparticles can improve the delivery of water-insoluble drugs, enhance the passage of chemotherapeutic agents across biological membranes [7], enable the drugs to be delivered only to the cancer cells [8], improve drug distribution, provide sustained release of a drug, and help in delivery of two or more drugs for combined therapy as compared with the non-encapsulated free drugs [9]. The nanoparticles’ (NPs) design parameters can be optimized to maximize their performances by modifying their composition, particle size, morphology, and surface properties, to increase the efficacy of treatment, reduce side effects, and overcome drug resistance. In this scenario, the development of nanoparticulate-based drug delivery systems holds promise for cancer therapy, because natural compounds fabricated at the nanometer scale exhibit drastically altered bioactivities and toxicity. Here, we discuss the recent updates on how low bioavailability, a major concern with these selected phytochemicals—apigenin, resveratrol (RES), curcumin (Cur), epigallocatechin-3-gallate (EGCG), 6-gingerol (6G), and quercetin (Qc)—is being circumvented by synthesizing as nanoformulations. In addition, we summarize several updates on preparation methods of NPs.

## 2. Advances in Utilizing NPs in Cancer Management and Therapy

With the increasing applications of NPs in cancer treatment and management, research on green-synthesized NPs has become a hotspot [10]. The conventional methods of generating NPs often require the use of aggressive chemical reducing agents, which are costly, complex, and can produce very toxic side-products. Thus, more cost-effective and environmentally benign alternative approaches which can minimize the risk of contaminating the environment are unequivocally needed. In this section, we summarize several NPs which are currently being employed for anticancer therapies and discuss their latest synthetic methods and applications in cancer management.

### 2.1. Gold NPs (AuNPs)

Au was utilized in infection treatment thousands of years ago in ancient India and China, and the applications of Au boom up with the development of nanotechnology. AuNPs possess low cytotoxicity, high surface area to volume ratio and stability, which support them as good candidates for chemotherapy and immunotherapy in cancer treatment [11]. AuNPs also gained attention as an ideal contrasting agent in X-ray and CT and emerged as a radiosensitizer for cancer early detection, diagnostics, and treatment [12,13]. In addition, several studies indicated the intrinsic antitumor property of AuNPs which can selectively kill cancer cells [14]. AuNPs have caught the attention of scientists for their use as drug carriers, and consequently their simpler synthesis via green chemistry has also become foremost importance. In the past, AuNPs were always synthesized by utilizing chemicals and solvents, which has a negative effect on environment and human health [15]. Recently, AuNPs synthesized by employing active compounds from plants are reported to be non-toxic towards normal human cells [16]. Moreover, the synthesis process is simple and environmentally friendly [17].

AuNPs synthesized by using Cur (Cur–AuNPs) are the most studied phytofabricated AuNPs so far (Figure 1). It has been confirmed that Cur is a well-performing reducing agent to produce functionalized AuNPs and maintains the stability upon conjugation with AuNPs [18,19]. In addition, Cur–AuNPs display dominant results in antioxidant properties compared with free Cur and regulate the cancer immunity to some extent. In vitro experiment of Cur–AuNPs also showed stronger competence in inducing apoptosis in the colon (HCT-116) and breast (MCF-7) human cancer cell lines than the free Cur and the 50% inhibiting concentration (IC_50_) of MCF-7 and MDA-MB 231 cell lines were found at 10.0 μM [20,21]. Some other phytochemicals can also be involved in reduction of metal ions to AuNPs in a one-step and eco-friendly synthetic process. Previous studies on AuNPs synthesized with quercetin as reducing agent (Qc–AuNPs) revealed that Qc–AuNPs inhibit the epithelial–mesenchymal transition, angiogenesis, and invasiveness via epidermal growth factor receptor (EGFR)/vascular endothelial growth factor receptor (VEGFR)-2-mediated pathway and induce apoptosis via inhibiting EGFR/phosphatidylinositol 3-kinase (PI3K)/protein kinase B (Akt)-mediated pathway in breast cancer cell lines (MCF-7 and MDA-MB 231) [22,23]. The underlying mechanism demonstrated that Qc–AuNP is a facile and economic-promising nanodrug with certain anticancer efficacy. Similarly, the AuNPs synthesized with EGCG also induced anticancer efficacy and tumor tissue selectivity [24,25]. To dig out the huge potential of natural anticancer substance, permutation and combination of numerous phytochemicals and AuNPs were conducted. While various ideal/positive outcomes are exciting, some inherent properties of the nanoparticle should be noted like the various shapes of AuNPs, as different particle shape and surface chemistry can affect the cytotoxicity and cellular upatake of metallic nanorods and nanospheres [26].

### 2.2. Silver NPs (AgNPs) 

Similar to AuNPs, AgNPs are also exhibiting antibacterial, antimicrobial, and anticancer activities with extensive uses in medicine [27]. Compared to Au, Ag is a more accessible metallic material for cancer treatment. Researchers are enthusiastic in green synthesis, and it is shown that plant extracts as both reducing and stabilizing agents can be adapted to synthesize AgNPs [28]. Green synthesis of AgNPs using phytochemicals has several advantages such as cost-effectiveness, eco-friendliness, and biocompatibility [29]. Different from the green synthesis of AuNPs, the research of AgNPs emphasize the synthesis of AgNPs itself, namely plant extract is usually considered as the reducing agent in the synthesis of AgNPs but is not concerned about the bioactive compounds present in the end-product. AgNPs without combining phytochemicals still show excellent anticancer properties through green synthesis. For example, there was no obvious evidence confirming the phytochemicals contributed the anticancer effect of the AgNPs synthesized by *Piper longum* leaf extracts in Hep-2 cell lines [30]. However, the role of phytochemicals in AgNPs should not be ignored as some clues indicate the bioactive compound could bind to the end-product AgNPs [31]. Though the anticancer properties of AgNPs have been proven, the composition of AgNPs influences the stability, interaction, and toxicity, etc. As for the phytochemicals conjugated AgNPs, besides the anticancer efficacy, the interaction between AgNPs and serum protein is interesting. Both Cur and EGCG-coated AgNPs can be conjugated on serum protein as a loose complex, which turns out to be a stable and potential biosensor [32]. Each element in this complex has cancer therapeutic ability.

### 2.3. Other Metal NPs

Other metals like copper and platinum are also adapted as raw materials to construct the phytochemical-inert metal NPs [33]. However, most of these products lack inspiring findings on the respective cytotoxicity effect. The nanotoxicity of metal NPs should be noted even though some researchers claim that there is negligible nanotoxicity for metal NPs. Nevertheless, the application of metal NPs should be cautious in vivo, as the toxicity especially for the Au and Ag mainly results from tissue accumulation, so the long-term toxicity should be considered seriously. 

### 2.4. Plant-Derived Edible NPs

Previous studies suggest that nanosized edible particles from plant cells may be exosome-like, which could serve interspecies communication roles, exert anti-inflammatory properties, and function against cancers [34,35]. In comparison with exosome derived from mammalian cells, plant-derived edible NPs show an economic advantage in scaling up for mass production [36]. As it is well known, biocompatibility and safety are the major barriers between laboratory and clinic in nanomedicine. Plant-derived edible NPs exhibit an unique advantage in these aspects, as they are consisting of high levels of lipids, few proteins, RNAs, which made them one of the safest therapeutic NPs [37]. Ginger is an ideal natural source to gain edible NPs. Zhang et al. found that the NPs isolated from ginger with abundant 6-gingerol (6G) and 6-shogaol, perform ideal stability, tissue selectivity, anti-inflammatory effect, and cancer therapy potential in mice [38]. It should be reminded that NPs by oral administration can also distribute into other organs by the circulation rather than simply binding to gastrointestinal tract cells. These NPs can protect the liver from alcoholic damage in mice, which is significant for hepatocellular carcinoma. Besides, the edible NPs isolated from *Citrus limon* are efficient in the inhibition of the growth of tumor in the leukemia mice model [39] (Figure 2). It should also be noted that harvesting of edible NPs in high yield and quality is difficult. However, recent extraction and purification techniques have shown promise, such as isosmotic buoyant density and isosmotic cushion ultracentrifugation, equilibrium density gradient ultracentrifugation, and differential ultracentrifugation plus density gradient centrifugation [40].

### 2.5. Plant Lipid-Derived NPs

Lipid NPs generated from edible plants can be used as nanocarriers of chemical drugs other than their inherent compounds. Using sonication, lipids extracted from plants can form nanostructures in which chemotherapeutic agents or phytochemicals can be embedded [41]. Lipid NPs are easily biodegradable and without biohazards to the environment, representing a novel and natural delivery system. Lipid NPs isolated from plants can deliver drugs to a specific location of the human body [42]. The most studied plant is grapefruit, and the grapefruit-derived lipid NPs have been demonstrated to deliver therapeutic agents in mouse CT26 and human SW620 colon cancer models [43] (Figure 3). Research shows ginger has a high proportion of lipid [44]. Data from the literature shows that lipid-derived NPs loaded with chemical drugs also display excellent advantages over artificial NPs in cancer therapy. Delivering doxorubicin by lipid-derived NPs had more efficiency than free doxorubicin [45]. With the process of geno-therapy, the cancer-suppression effect of siRNA has become a hot area. The natural lipid carriers loaded with the miRNA-18a also performed well in inhibition of liver cancer metastasis to normal tissues [46]. The natural lipid NPs are also considered as an ideal carrier for siRNA delivery. Loading of CD98-siRNA into lipid NPs can target it specifically to gut tissues by oral administration, reduce the expression of CD98 and show promise for immunoregulation [47].

Though it is a promising approach for drug delivery to utilize NPs derived from plant, more details should be noted as the complex constituents and interaction of the plant bioactive and therapeutic agents are not negligible. As drug-metabolizing enzymes have a great influence on pharmacokinetics, a large amount of therapeutic agents, including chemical drugs and plant constituents, can act as agonist or inhibitor of these enzymes. Thus, consideration should be undertaken on whether the metabolizing processes would be changed when the drug-loaded NPs are established with plant and chemicals drugs. For example, the furanocoumarins abundant in grapefruit are potent inhibitors of cytochrome P450s, important enzyme families in drug metabolism [48]. If drug-loaded NPs are constructed by grapefruit, it is likely to influent the drug’s plasma concentration and bioavailability. A case in point is the combination of the grapefruit component with tyrosine kinase inhibitors (e.g., erlotinib, nilotinib), which could increase the risk of adverse reaction like Torsades de pointe or bone marrow suppression for the persistent higher plasma drug concentration. Similar effects also exist in many other plants and drugs like Saint-John’s Wort, ginseng, paclitaxel, etc. [49]. As a result, the interaction of the constituents should be taken seriously to avoid aggravating side effects and optimizing the administration strategies. We hereby attempt to perform a SWOT (strengths, weaknesses, opportunities, and threats) analysis of the phytofabricated NPs (Figure 4).

## 3. Phytochemicals Conjugated with NPs as Nanomedicine

### 3.1. Apigenin

Apigenin (4′, 5, 7,-trihydroxyflavone) is a popular member of flavones because of its low intrinsic toxicity and striking oxidation resistance, anti-inflammatory, and anti-carcinogenic properties [50]. Apigenin exerts its anticancer effect by inducing cell apoptosis and autophagy, modulating cell cycle, inhibiting cell migration and invasion, and induction of immune responses [51]. Studies assayed in head and neck squamous cell carcinoma, glioblastoma cells, and triple-negative breast cancer cells respectively demonstrate that apigenin shows selective cell cytotoxicity to cancer stem cells which are closely associated with cell proliferation, metastasis, and drug resistance of cancer [52,53,54]. However, according to the biopharmaceutics classification system, apigenin was categorized as a Class II drug because of its low solubility which greatly holds back its use in clinical settings [55]. Hence, in order to attain a better bioavailability, a new formulation is necessary.

In a study, apigenin was loaded with poly(lactic-co-glycolide acid) (PLGA) NPs and the anti-carcinogenic effect was evaluated in benzo[*a*]pyrene and ultraviolet-B induced mouse skin cancer model. Results demonstrated that it showed better effects than free apigenin by reducing tissue damage and frequency of chromosomal aberrations and inducing mitochondrial apoptosis [56]. In another study, an apigenin–phospholipid phytosome (APLC) was designed to improve solubility, dissolution, in vivo bioavailability, and antioxidant potential of apigenin. Compared with pure apigenin, APLC was found to be over 36-fold higher than that of the aqueous solubility of apigenin, and improved the oral bioavailability and restoration of all carbon tetrachloride-elevated rat liver function marker enzymes [57]. Apigenin NPs produced by the liquid antisolvent precipitation technique was assayed in a rat model. The experimental results showed that the solubility, dissolution rates, oral bioavailability, and antitumor effect of the apigenin NPs are higher than the raw apigenin. In addition, there is no toxic effect on the organs of rats [58].

### 3.2. Resveratrol (RES)

Resveratrol (3,4′,5-trihydroxy-trans-stilbene), a non-flavonoid polyphenol, is a phytoestrogen that attracts significant attention from researchers for its potent effects of anti-oxidant, anti-inflammatory, and anticancer properties in many cancers [59,60,61]. In addition, it is more striking in reversing drug resistance and the sensitizing of cancer cells for chemotherapy and radiotherapy [62,63]. However, it is a pity to know that the poor water solubility and rapid metabolism of RES in the intestine and liver results in low bioavailability of less than 1% which hinders its pharmacological potential [64,65,66]. In this scenario, the development of nano-engineered systems to solve this problem is needed. In a study, non-small cell lung carcinoma cells were studied in Swiss albino mice after the administration of gelatin NPs-loaded RES (RES–GNPs). Research found that RES–GNPs enhanced anticancer efficacy of RES by inducing cell cycle arrest in the G_0_/G_1_ phase [67]. In another work, a transferrin-targeted, RES-loaded liposome (Tf–RES–L) was produced to treat glioblastoma. Since transferrin is up-regulated in glioblastoma, this makes Tf–RES–L able to be site-specifically targeted. In vitro experiments showed the stability, excellent drug loading capability and prolonged drug-release time of Tf–RES–L while in vivo studies exerted better anticancer effect and higher survival rate in mice [68]. Similarly, in prostate cancer cells, RES-loaded PLGA mediated programmed cell death by promoting cell arrest at G_1_/S phase of cell cycle [69]. Moreover, RES was loaded with AuNPs (RES–AuNPs) and the anti-hepatoma efficacy was evaluated in vitro and in vivo. RES–AuNPs was proven to remarkably inhibit tumor growth, promote tumor apoptosis and decrease the expression of vascular endothelial growth factor (VEGF) in xenograft studies. Both in vitro and in vivo studies showed ameliorative potential in antitumor effects compared to pure RES which may be due to the higher concentration of RES in mitochondria with the help of AuNPs [70].

### 3.3. Curcumin (Cur)

Curcumin (1,7-bis(4-hydroxy-3-methoxyphenyl)-1,6-heptadiene-3,5-dione) is a bioactive compound extracted from the root of the turmeric plant. It has been used in medical treatment for centuries because of its anti-oxidative, anti-inflammatory, analgesic, antiseptic, and antimalarial properties [71,72,73]. Cur is famous throughout these years as it can be a potent chemosensitizer against chemoresistance by modulating multiple cell signaling pathways and cause cell apoptosis [74,75,76]. Nonetheless, numerous formulations need to be done to improve its low bioavailability caused by low water solubility, low stability, poor absorption, and rapid systemic metabolism [77,78,79]. Since it is known that combinatorial strategies can usually obtain an optimized effect, a co-delivery system of doxorubicin and Cur in pH-sensitive NPs was constituted with an amphiphilic poly copolymer. The experiments in human liver cancer SMMC 7721 cells exerted enhanced cellular internalization of drugs and a high rate of cancer cell apoptosis [80]. Similarly, a single walled carbon nanotubes-based drug delivery system which was modified with alginate and chitosan got the same result in human lung adenocarcinoma (A549) cells [81]. Furthermore, researchers created an ideal delivery system of Cur by loading it with D,L-PLGA NPs and coated it with chitosan and PEG to obtain an optimum therapeutic effect. In vitro, cellular studies reveal that the migration and invasion ability of metastatic pancreatic cancer are reduced which means that the novel formulation revealed superior cytotoxicity and apoptosis-inducing ability [82].

### 3.4. (−)-Epigallocatechin-3-gallate (EGCG)

EGCG ((−)-cis-3,3′,4′,5,5′,7-hexahydroxy-flavane-3-gallate) is the most abundant phytochemical in green tea, which influences the proliferation, growth, and metastasis of tumors. Research evidence from in vitro and in vivo studies demonstrate the potential of this particular natural compound in preventing or interfering with many forms of cancers, which include but are not limited to skin and lung cancers. Nevertheless, these in vitro and in vivo encouraging results have not been reproduced in the clinic. This is mostly due to its short half-life and poor systemic absorption resulting in low bioavailability.

Fortunately, the utilization of nanotechnology can improve the pharmacokinetic and pharmacodynamic profiles of conventional therapeutic formulations. A study with EGCG-loaded lipid nanocapsules was performed to enhance its stability in the plasma [83]. Different approaches have been used to improve the bioavailability by encapsulating EGCG with polylactic acid and polyethylene glycol (PLA–PEG) NPs [84]. PLA–PEG NPs encapsulated EGCG retained its biological activity with over a ten-fold dose advantage in exerting anticancer effects in vitro in 22Rν1 prostate carcinoma cell line as well as in vivo in athymic nude mice implanted with human prostate cancer cells.

A group studied excellent anti-proliferative and pro-apoptotic effects of EGCG-chitosan NPs on human melanoma in vitro and in vivo [85]. It was found that EGCG encapsulated in chitosan NPs, in comparison to native EGCG, showed a marked induction of apoptosis in Mel 928 human melanoma cells with about eight-fold better efficacy. Chitosan-based nano-formulation containing EGCG was also observed to have a substantial improvement of therapeutic benefit against melanoma tumors compared to the native agent in a mouse model of melanoma.

In a mouse melanoma tumor model, Sm^III^–EGCG nanocomplexes are directly compared with a clinical anticancer drug, 5-fluorouracil, and shows remarkable therapeutic effects on primary melanoma tumors and inhibition of metastasis of melanoma from invading other organs through targeted therapeutic effects [86]. Moreover, these results suggested that Sm^III^–EGCG complexes exhibited significantly lower adverse side effects than 5-fluorouracil when the anticancer therapy was performed on melanoma primary tumors. Similarly, in vivo results revealed that Sm^III^–EGCG nanocomplexes selectively and effectively induced the apoptosis of tumor cells with negligible effects on the normal healthy cell lines.

Recently, one study reported that self-assembly of the PEG–EGCG with Sunitinib (SU) leads to the formation of stable micellar nanocomplex (SU-MNC), which have greater anticancer effectiveness than conventional SU formulations on human renal cell carcinoma-xenografted mice [87]. When injected into mice, SU-MNC efficiently inhibited tumor growth with less systemic toxicity. Improved efficacy of SU-MNC was attributed to the carrier−drug synergies as well as tumor-targeted delivery. This study shows that EGCG-based nanocarriers would provide a more effective and safer strategy for cancer therapy, suggesting an opportunity for potential improvement in therapeutic efficacy of the nanocarrier platform.

### 3.5. 6-Gingerol (6G)

6-Gingerol (1-(4′-hydroxy-3′-methoxyphenyl)-5-hydroxy-3-decanone), one of the important natural compounds isolated from the rhizome of ginger, has captured a lot of research interests due to its wide range of biological activities like anti-inflammatory, antitumor, and so forth [88]. Gingerol contains a series of phytoconstituents including 4-gingerol, 6-gingerol, 8-gingerol, 10-gingerol, and 12-gingerol, with 6-gingerol (6G) being the pharmacologically active component [89]. Unfortunately, its application is limited in the clinical settings mainly due to poor water solubility, temperature, pH, and oxygen sensitivity and light instability. In order to deliver 6G in a targeted and controlled manner and achieve better clinical efficacy, many new drug delivery systems have been studied and developed in recent years, of which nano-drug delivery system has become a topic of rapidly expanding scientific interest.

Manatunga et al. [90] reported a novel pH sensitive sodium alginate, hydroxyapatite bilayer-coated iron oxide nanoparticle composite (IONP/HAp-NaAlg) loaded with two anticancer drugs, 6G or Cur. The prepared nanocarrier system has shown the higher encapsulation efficiency and sustained pH-controlled releasing ability, which make sure the release of 6G or Cur in a targeted and controlled manner to treat cancer.

Loading of 6G into nanosized proliposomes has recently been studied for anticancer efficacy in SD rat and it has been shown that the formulation of 6G reduced clearance by macrophage thus prolonged the blood circulation time [91]. Encapsulation of 6G in nanosized proliposomes was also tested in HepG-2 cells that demonstrated that the antitumor effect of 6G was improved by its entrapment in proliposomes.

6G-loaded nanostructured lipid carriers have been shown to improve the water solubility and the oral bioavailability of 6G [92]. These composite NPs were observed to have a suitable size distribution, drug encapsulation efficiency, and drug release kinetics. In another important study, the toxic effects of encapsulation of 6G in PEGylated nanoniosome was tested in the breast cancer cell line T47D and the study exhibited that IC_50_ of the nanoformulation is less than the standard drug [93]. 6G-loaded PEGylated nanoniosome have also been shown to display more stability and slower release of the compound. Another characteristic feature of PEGylated nanoniosome noted was its stability during storage and the capacity of drug loading.

Finally, magnetic hydroxyapatite (m-HAP) NPs conjugated to 6G were tested in MCF-7 cells and HepG-2 cells, and it has been shown to display more effective inhibition of the proliferation of cancer cells than 6G alone [94]. Another characteristic feature of m-HAP NPs noted was considerably higher loading capacity and reduced toxic effects on non-targeted, non-cancerous cells.

### 3.6. Quercetin (Qc)

Quercetin (3,3′,4′,5,7-pentahydroxyflavone), an attractive polyphenolic active compound, which is found in several dietary plant foods such as apples, onion, and red grapes, has been proven to possess a variety of pharmacological benefits [95,96]. A large number of studies conducted over the past years have shown that this particular natural compound can impede one or more steps in carcinogenesis in various cancer cell lines [97,98]. Additionally, it has been previously demonstrated that Qc in combination with chemotherapeutic drugs maximizes the efficacy of these agents in induction of apoptosis in cancer cells and is very effective in the elimination of multi-drug resistance [99,100]. In spite of this beneficial effect, the use of Qc in clinical application met with limited success. Therefore, current studies are focused on the development of nanoformulations which would overcome low aqueous solubility and poor chemical stability of natural Qc.

Minaei et al. [101] prepared composite NPs by mixing Qc and lecithin and examined its potential use in doxorubicin-induced apoptosis. The data from this study demonstrated that combination of nano-Qc and doxorubicin increased toxic effects of doxorubicin in human MCF-7 breast cancer cells. Furthermore, the formulation provided improved drug loading, sustained, and sequential release of both agents.

In another study, a near-infrared-responsive drug system-based on Au nanocages with Biotin–PEG–SH modification was synthesized for the combination of doxorubicin and Qc [102]. In this study, the resultant nanocomplex was exhibited to have much more potent effects on MCF-7/ADR cells growth inhibition under near-infrared irradiation. In addition, the co-delivery of doxorubicin and Qc could effectively increase the intracellular accumulation of doxorubicin and distribution in nuclei.

The anti-proliferative and pro-apoptotic effects of lipid–polymeric NPs (LPNs)-loaded with vincristine and Qc on human lymphoma have been studied both in vitro and in vivo [103]. To be noticed, the lethality of treated Raji/VCR cells was higher than that of the free drugs. These LPNs were confirmed to completely inhibit tumor growth along with a lower toxicity in a mouse model of lymphoma. Co-encapsulation of vincristine and Qc in the same LPNs can combine the efficiency of these two drugs and bring about synergistic effect and has potential as a novel therapeutic approach to overcome chemo-resistant lymphoma.

Lastly, in another study, the anticancer potential of nanoparticle of Qc alone and in combination with cisplatin nanoparticle (LPC) was studied in a bladder carcinoma model and the study suggested that the nanoformulation yielded significantly enhanced antitumor efficiency in combination with LPC [104]. Encapsulated polyphenol in lipid calcium phosphate NPs protected Qc from degradation, facilitated increased accumulation at the tumor site through enhancing the drug permeability, and enhanced the tumor penetration of second-wave NPs administered. For the sake of simplicity, we summarized and presented the information of this section in Table 1.

## 4. Conclusions

Abundant natural phytochemicals are potential anticancer drugs. It is widely accepted that nanotechnology could be a future direction in cancer treatment. The application of phytochemicals in combination with nanotechnology amplifies the therapeutic effect and provides a new way to solve the difficult economic and environmental problems of nanotechnology. Therefore, combining phytochemicals with nanotechnology is a promising approach. However, challenges of nanotechnology are not yet fully settled. Despite research advancements in this area with various modifications on the nanocarrier platform to improve pharmaceutical properties of therapeutic molecules, achieving desirable effectiveness still remains an issue for clinical success. One of the major drawbacks is in general these NPs can only encapsulate small amounts of compounds. Though tailor-made nanomaterials functionalized with specific ligands could enable loaded drugs to function at lower doses, their improved efficacy over conventional drugs has remained marginal. The main reason is that nanocarriers are just excipients for delivering drugs and not therapeutically active. To overcome the obstacles, more innovative nanocarriers (for example, phytofabricated NPs) that have inherent therapeutic properties need to be developed. In addition, it is important to point out that the safety of nanocarriers remains largely unexplored, as nanomaterials may not have immediate health impact.

Furthermore, the manufacturing of nanomedicinal products for commercialization is a key obstacle. The determination of optimal physicochemical parameters of NPs is needed. As the involvement of multiple steps or complicated technologies for the production of NPs, the use of well-designed manufacturing processes is essential. What is more, the clinical benefit must be guaranteed, as the manufacturing cost is generally high. Indeed, large scale-production is technically challenging, the transition from laboratory to clinic is nearly always accompanied by the optimization of formulation parameters. Even a subtle change in the formulation methods then the physicochemical properties of NPs may vary from batch to batch. Suffice to say, we still have a journey in the large-scale production of nanoparticles for drug delivery and many challenges to be conquered.

## Figures and Tables

**Figure 1 molecules-24-04246-f001:**
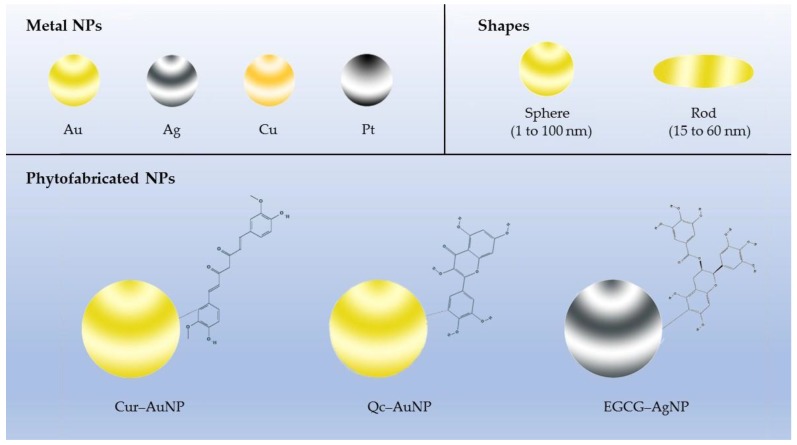
Phytofabricated metal nanoparticles (NPs) synthesized by employing active compounds from plants (e.g., curcumin, quercetin, and epigallocatechin-3-gallate (EGCG)) as reducing agent. The synthetic process is simple and environmentally friendly. The Au nanospheres have a diameter between 1 and 100 nm and the length of Au nanorods range from 15 to 60 nm.

**Figure 2 molecules-24-04246-f002:**
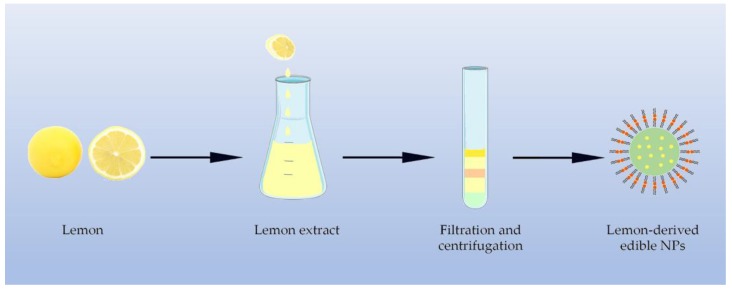
Exosome-like edible nanoparticles were isolated from representative plant (e.g., lemon). Edible nanoparticles can be isolated from plants using filtration and centrifugation.

**Figure 3 molecules-24-04246-f003:**
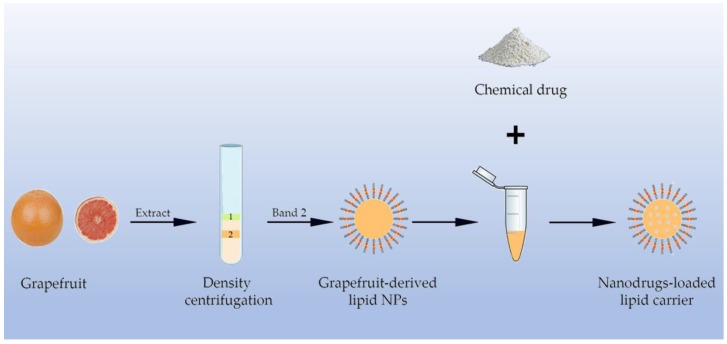
Lipid NPs were isolated from plants (e.g., grapefruit), and then phytochemicals were embedded inside them.

**Figure 4 molecules-24-04246-f004:**
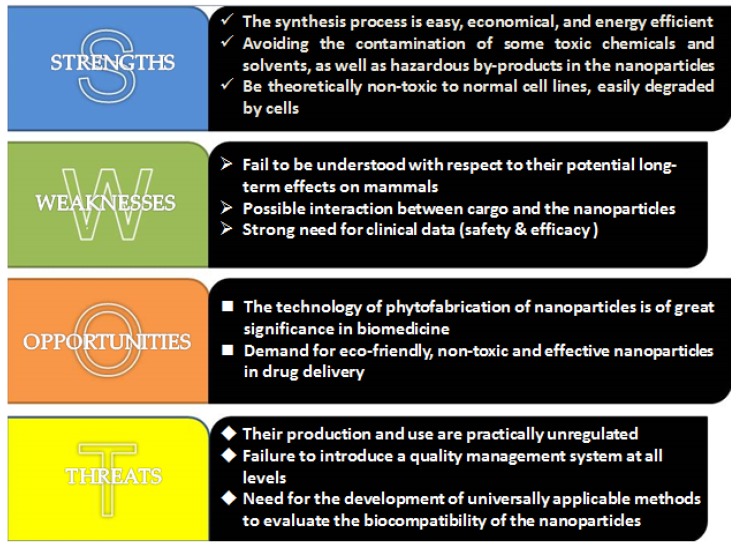
The strengths, weaknesses, opportunities, and threats (SWOT) analysis of phytofabricated NPs from our point of view.

**Table 1 molecules-24-04246-t001:** Studies employing nanotechnology for delivery of (phyto)chemicals.

(Phyto)chemicals	Studying Group	Nanoparticles	Cancer Types	Conditions	Efficacy as Compared with Free Forms	Reference
**Apigenin**	Das et al. (2013)	PLGA	Skin cancer	In vitro and in vivo	Enhanced anti-carcinogenic effect	[56]
	Telange et al. (2017)	APLC	Liver cancer	In vitro and in vivo	Improved aqueous solubility, dissolution, in vivo bioavailability, and antioxidant activity	[57]
	Wu et al. (2017)	Liposomes	Hepatoma	In vitro and in vivo	Improved solubility and bioavailability	[58]
**Resveratrol**	Karthikeyan et al. (2015)	Gelatin	Lung cancer	In vitro	Better stability; improved drug- loading capacity; sustained drug-release; improved cytotoxicity	[67]
	Jhaveri et al. (2018)	Liposomes	Glioblastoma	In vitro and in vivo	Enhanced solubility and stability; sustained drug-release; better tumor selectivity	[68]
	Nassir et al. (2018)	PLGA	Prostate cancer	In vitro	Enhanced anti-carcinogenic effect by inducing mitochondrial-dependent apoptosis and cell arrest	[69]
	Zhang et al. (2019)	Au	Hepatoma	In vitro and in vivo	Inhibition of tumor growth; induced tumor apoptosis and decreased the expression of VEGF	[70]
**Curcumin–Doxorubicin**	Zhang et al. (2017)	pH-sensitive nanoparticles	Liver cancer	In vitro and in vivo	Low polydispersity and high encapsulation efficiency; enhanced release in the acidic environment; inhibition of angiogenesis	[80]
**Curcumin**	Singh et al. (2018)	Single walled carbon nanotubes	Lung adenocarcinoma	In vitro	Improved aqueous solubility; a moderate and ideal drug delivery system; enhanced anticancer effect	[81]
	Arya et al. (2018)	PLGA	Metastatic pancreatic cancer	In vitro	Superior cytotoxicity; enhanced anti-migratory; anti-invasive and apoptosis-inducing ability	[82]
**EGCG**	Siddiqui et al. (2010)	PLA–PEG	Prostate cancer	In vitro and in vivo	Enhanced bioavailability; superior inhibition of angiogenesis	[84]
	Siddiqui et al. (2014)	Chitosan	Melanoma	In vitro and in vivo	Excellent anti-proliferation	[85]
	Li et al. (2019)	Sm^III^ nanocomplexes	Metastatic melanoma	In vitro and in vivo	Decreased viability; inhibition of wound-induced migration; prevention of metastatic lung melanoma from spreading	[86]
**EGCG–Sunitinib**	Yongvongsoontorn et al. (2019)	MNC	Renal carcinoma	In vitro and in vivo	Enhanced anticancer effects and less toxicity; inhibition of angiogenesis	[87]
**6-Gingerol/Curcumin**	Manatunga et al. (2017)	IONP/HAp-NaAlg	Breast cancer	In vitro	Targeted and controlled release over a period of time	[90]
**6-Gingerol**	Wang et al. (2018)	Nanosized proliposomes	Liver cancer	In vitro and in vivo	Improved water solubility; sustained drug release; enhanced oral bioavailability	[91]
	Wei et al. (2018)	Lipid nanocapsules	Liver cancer	In vitro	Better stability and slower drug release; targeted delivery	[92]
	Behroozeh et al. (2018)	PEGylated nanoniosome	Breast cancer	In vitro and in vivo	Enhanced bioavailability	[93]
	Manatunga et al. (2018)	m-HAP	Breast and liver cancers	In vitro	Increased stability; controlled and targeted delivery; minimizing toxicity	[94]
**Quercetin–Doxorubicin**	Minaei et al. (2016)	Lecithin	Breast cancer	In vitro and in vivo	Elevated efficacy of chemotherapeutics by increasing the permeability of tumor cells to chemical agents	[101]
	Zhang et al. (2018)	Au nanocages	Breast cancer	In vitro and in vivo	Inhibition of tumor growth	[102]
**Quercetin–Vincristine**	Zhu et al. (2017)	Lipid-polymeric	Lymphoma	In vitro and in vivo	Improved bioavailability and metabolic stability; remodeled tumor microenvironment and increased the penetration of second-wave nanoparticles into the tumor nests	[103]
**Quercetin–Cisplatin**	Hu et al. (2017)	Lipid calcium phosphate	Bladder carcinoma	In vitro and in vivo	Enhanced permeation and retention effect; selective targeting; greater antitumor efficacy and minimized toxicity	[104]

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
