# Peer review of "Phytofabrication of Nanoparticles as Novel Drugs for Anticancer Applications"

_molecules, 2019, doi:10.3390/molecules24234246_

Round 1
Reviewer 1 Report
This review article refers to the design and properties of nanoparticules containing phytochemicals and their potential use as anticancer agents. This is not a new topic; there are several recent review on this topic. They described the fabrication of nanoparticules, based on gold and silver essentially, that include different types of phytochemical products, principally apigenin, resveratrol, curcumin, EGCG and gingerol. These are the most classical phytochemicals, again not very original or inspiring. The literature survey is OK but I found this manuscript of limited interest. I didn’t learn anything new. The following specific comments can be made:
The manuscript is very poorly illustrated. Figs 1-3 are trivial, with NPs shown as ordinary spheres, without details about their architecture or types or properties (drug inside? drug-coating? Size, structure, etc). A real effort should be made to improve the manuscript graphically (there is no need to show an Eppendorf tube !). A critical analysis of the advantage/disadvantages of NPs made with phytochemicals is lacking. I will recommend to construct a kind of SWOT analysis to indicate the positive and negative points, as well as the opportunities. One of the key limitations is that in general these NPs can only encapsulate small amounts of compounds, and for this reason it works well only with high potent drugs (such as doxorubicin), but it can be a major issue with low potency phytochemicals. The active concentration can be difficult to reach when using NPs with low potency molecules. Please comment. Manufacturing at a high-scale of NPs is a problem. This is briefly evoked in the conclusion. It certainly warrants a specific paragraph and discussion. Table 1 provides a collection of drug/nanoparticules and cite their (in vitro) effects against on cancer cells. It would be useful to comment further this table and suggest new directions of research. The English language need to be revised in some places. There are sentences very unclear in some cases.For these reasons, I cannot recommend the publication of this manuscript in the present form. An extensive revision is needed.
Author Response
Reviewer: 1
Comments to the Author
This review article refers to the design and properties of nanoparticules containing phytochemicals and their potential use as anticancer agents. This is not a new topic; there are several recent review on this topic. They described the fabrication of nanoparticules, based on gold and silver essentially, that include different types of phytochemical products, principally apigenin, resveratrol, curcumin, EGCG and gingerol. These are the most classical phytochemicals, again not very original or inspiring. The literature survey is OK but I found this manuscript of limited interest. I didn’t learn anything new. The following specific comments can be made.
Our response: Thank you very much for referee 1’s comments and constructive suggestions. We have addressed all the concerns and listed point by point as below:
1) This is not a new topic; there are several recent review on this topic.
Our response: Thank you for your suggestion. Although phytochemicals mentioned in the review are typical, we are focusing on not only the latest applications of nanoparticles in combination with phytochemicals as nanomedicine, but also the use of various nanocarriers in delivering the (phyto)chemicals.
2) Figs 1-3 are trivial, with NPs shown as ordinary spheres, without details about their architecture or types or properties (drug inside? drug-coating? Size, structure, etc). A real effort should be made to improve the manuscript graphically (there is no need to show an Eppendorf tube !).
Our response: Thank you for your suggestions. We have added more details in the revised figures, but we are sorry that we cannot cover every aspect you advised. For the figure 1, though various structures of the golden nanoparticle like nanospheres, nanorods, nanoshells and nanocages are studied, we only show the sphere and rod in the revised figure for these reasons: 1) the sphere is a representative and most-research green synthesis or phytochemical conjunction metal nanostructure. 2) Nanorod is also an ideal structure preferred by some researcher for the possible higher efficiency. 3) The nanoshell and nanocage are also promising but they are applied more in cooperation with other agents like therapeutic RNA or polymer, and there are no convincing studies in these two structures with phytochemical. As for the size, we have added some information. Additionally, we find that most research attach importance to the efficacy of the nanoparticle rather than how the phytochemical agents combine to the metal nanoparticle, so limited resource can be consulted. We have added the ideograph to show the combination in the revision. For the other two figures, a more detailed schematic representation was shown. As the information on the extraction or contribution of the nanoparticle from the plant are accessible in the original article, the readers can get very detailed parameters by referring to those research articles. We therefore in here only show the general and more simplified processes in our revision.
3) A critical analysis of the advantage/disadvantages of NPs made with phytochemicals is lacking. I will recommend to construct a kind of SWOT analysis to indicate the positive and negative points, as well as the opportunities.
Our response: We agree with you on this point. In the revised manuscript, we have included a new figure 4 of SWOT (strengths, weaknesses, opportunities and threats) analysis of phytofabricated NPs.
4) One of the key limitations is that in general these NPs can only encapsulate small amounts of compounds, and for this reason it works well only with high potent drugs (such as doxorubicin), but it can be a major issue with low potency phytochemicals. The active concentration can be difficult to reach when using NPs with low potency molecules. Please comment.
Our response: We agree with you on this point. In the revised manuscript, we have commented this limitation.
5) Manufacturing at a high-scale of NPs is a problem. This is briefly evoked in the conclusion. It certainly warrants a specific paragraph and discussion.
Our response: Thank you for your suggestion. In the revised manuscript, we have discussed the problem of manufacturing NPs at a high-scale in detail.
6) Table 1 provides a collection of drug/nanoparticules and cite their (in vitro) effects against on cancer cells. It would be useful to comment further this table and suggest new directions of research.
Our response: We agree with you on this point. In the revised manuscript, we have added two columns respectively for in vitro or in vivo experiment and the improvement of efficacy with nanotechnology used when compared with their free forms.
7) The English language need to be revised in some places. There are sentences very unclear in some cases.
Our response: We apologize for this. In the revised manuscript, we have performed thorough language editing and carefully checked the entire manuscript staring with the title.
Editor's Comments to Author:
Editor: Ángela Toribio, M.Sc.
Comments to the Author:
(There are no comments.)
Our response: Thank you very much for editor’s positive feedback on our manuscript.
Overall changes in the revised manuscript
For the text, it has been revised extensively according to the comments from both reviewers. New references have been added and irrelevant references removed, and cited accordingly in all sections. All the possible English misusage, unfriendly mode of writing, and possible typos have been corrected. All changes have been tracked and indicated on the right of each page.
For the figures, the original Figures 1, 2 and 3 were revised, and one new additional figure showing the SWOT analysis was added as figure 4 in the revised manuscript.
For the table, we have added two columns respectively for in vitro or in vivo experiment and the improvement of efficacy with nanotechnology used when compared with their free forms.
Other major changes in the revised manuscript:
- The number of references, from a total of 103 in the original submission, after adding/removing new and old references, the total number is 104 now in the revised manuscript.
We hope that you will find our revised manuscript with satisfaction and thank you for your time for reviewing it.
Reviewer 2 Report
This manuscript gives an overview on the application of nanotechnology in the naturally occurring products. It would help the readers an evident view on the improvement of efficacy with nanotechnology used when compared to their respective single agent therapy, if the a column of fold of improvement was added on to the Table 1. Moreover, a comment or perspective for the realistic future clinical trials for these combination therapy with nanoparticle formulation is recommended so that readers can have insight into how much would these combination be practical. Moreover, in consideration of grapefriut is not recommended to co-administrated with some particular drugs, therefore when the grapefruit phytoparticles are being used, any potential side effect could be induced ? This concern should be discussed or unravel in the relavent content.
Author Response
Reviewer: 2
Comments to the Author
This manuscript gives an overview on the application of nanotechnology in the naturally occurring products. It would help the readers an evident view on the improvement of efficacy with nanotechnology used when compared to their respective single agent therapy, if the a column of fold of improvement was added on to the Table 1. Moreover, a comment or perspective for the realistic future clinical trials for these combination therapy with nanoparticle formulation is recommended so that readers can have insight into how much would these combination be practical. Moreover, in consideration of grapefriut is not recommended to co-administrated with some particular drugs, therefore when the grapefruit phytoparticles are being used, any potential side effect could be induced ? This concern should be discussed or unravel in the relavent content.
Our response: Thank you very much for referee 2’s comments and constructive suggestions. We have addressed all the concerns and listed point by point as below.
1) It would help the readers an evident view on the improvement of efficacy with nanotechnology used when compared to their respective single agent therapy, if a column of fold of improvement was added on to the Table 1.
Our response: We agree with you on this point. In the revised manuscript, we have added two columns respectively for in vitro or in vivo experiment and the improvement of efficacy with nanotechnology used when compared with their free forms.
2) Moreover, a comment or perspective for the realistic future clinical trials for these combination therapy with nanoparticle formulation is recommended so that readers can have insight into how much would these combination be practical.
Our response: Thank you for your suggestion and we agree with you that it would be nice if we can include some data with respect to the clinical trials. However, it may be still impossible to give some perspective comments due to the fact that there is no information of the results of these clinical trials yet that have been performed on nanophytochemicals.
3) Moreover, in consideration of grapefriut is not recommended to co-administrated with some particular drugs, therefore when the grapefruit phytoparticles are being used, any potential side effect could be induced ? This concern should be discussed or unravel in the relavent content.
Our response: We agree with you on this point. However, we find that the previous studies did not attach much importance to the interplay on phyto-ingredient like grapefruit and the drug. It should be emphasized for the potential increasing risk of fatal adverse reaction. We have added more commentary on this perspective in the revised manuscript.
Editor's Comments to Author:
Editor: Ángela Toribio, M.Sc.
Comments to the Author:
(There are no comments.)
Our response: Thank you very much for editor’s positive feedback on our manuscript.
Overall changes in the revised manuscript
For the text, it has been revised extensively according to the comments from both reviewers. New references have been added and irrelevant references removed, and cited accordingly in all sections. All the possible English misusage, unfriendly mode of writing, and possible typos have been corrected. All changes have been tracked and indicated on the right of each page.
For the figures, the original Figures 1, 2 and 3 were revised, and one new additional figure showing the SWOT analysis was added as figure 4 in the revised manuscript.
For the table, we have added two columns respectively for in vitro or in vivo experiment and the improvement of efficacy with nanotechnology used when compared with their free forms.
Other major changes in the revised manuscript:
- The number of references, from a total of 103 in the original submission, after adding/removing new and old references, the total number is 104 now in the revised manuscript.
We hope that you will find our revised manuscript with satisfaction and thank you for your time for reviewing it.
Reviewer 3 Report
In this review, several research about nanoparticle conjugated with phytochemicals for anticancer applications was described and organized. Overall, the quality of article is good. But some points need to be improved.
Stability of NP in vivo is important for application. In this review, authors should put the informations of in vitro or in vivo experiment in Table 1. Listed phytochemcals without conjugated to NP have different and specific properties for anti-cancer properties. Authors can make these informations and references into a table. Authors can add comments to each parts, such as advantages and disadvantages, and usage restrictions of these NPs. Figure 1, 2, 3 could add more information.
Author Response
Reviewer: 3
Comments to the Author
In this review, several research about nanoparticle conjugated with phytochemicals for anticancer applications was described and organized. Overall, the quality of article is good. But some points need to be improved.
Stability of NP in vivo is important for application. In this review, authors should put the informations of in vitro or in vivo experiment in Table 1. Listed phytochemcals without conjugated to NP have different and specific properties for anti-cancer properties. Authors can make these informations and references into a table. Authors can add comments to each parts, such as advantages and disadvantages, and usage restrictions of these NPs. Figure 1, 2, 3 could add more information.
Our response: Thank you very much for referee 3’s comments and constructive suggestions. We have addressed all the concerns and listed point by point as below.
1) Stability of NP in vivo is important for application. In this review, authors should put the informations of in vitro or in vivo experiment in Table 1. Listed phytochemcals without conjugated to NP have different and specific properties for anti-cancer properties. Authors can make these informations and references into a table.
Our response: We agree with you on this point. In the revised manuscript, we have added two columns respectively for in vitro or in vivo experiment and the improvement of efficacy with nanotechnology used when compared with their free forms.
2) Authors can add comments to each parts, such as advantages and disadvantages, and usage restrictions of these NPs.
Our response: Thank you for your suggestion and we agree with you that it would be nice if we can add comments to each parts. In the revised manuscript, we have added more comments to the above issues.
3) Figure 1, 2, 3 could add more information.
Our response: Thank you for your suggestion. More details like the shapes and structures of nanoparticles have been added in the revised manuscript.
Editor's Comments to Author:
Editor: Ángela Toribio, M.Sc.
Comments to the Author:
(There are no comments.)
Our response: Thank you very much for editor’s positive feedback on our manuscript.
Overall changes in the revised manuscript
For the text, it has been revised extensively according to the comments from both reviewers. New references have been added and irrelevant references removed, and cited accordingly in all sections. All the possible English misusage, unfriendly mode of writing, and possible typos have been corrected. All changes have been tracked and indicated on the right of each page.
For the figures, the original Figures 1, 2 and 3 were revised, and one new additional figure showing the SWOT analysis was added as figure 4 in the revised manuscript.
For the table, we have added two columns respectively for in vitro or in vivo experiment and the improvement of efficacy with nanotechnology used when compared with their free forms.
Other major changes in the revised manuscript:
- The number of references, from a total of 103 in the original submission, after adding/removing new and old references, the total number is 104 now in the revised manuscript.
We hope that you will find our revised manuscript with satisfaction and thank you for your time for reviewing it.
Round 2
Reviewer 1 Report
The authors have addressed all my previous comments and made useful changes to improve the presentation and quality of this manuscript. The overall message is still not very new but the literature survey is appropriate and the comments are certainly useful to those working in the field of nanomedicine. I have no other restriction; the manuscript can be published.